# Effect of the Hybrid Assistive Limb on the Gait Pattern for Cerebral Palsy

**DOI:** 10.3390/medicina56120673

**Published:** 2020-12-07

**Authors:** Yuki Mataki, Hirotaka Mutsuzaki, Hiroshi Kamada, Ryoko Takeuchi, Shogo Nakagawa, Kenichi Yoshikawa, Kazushi Takahashi, Mayumi Kuroda, Nobuaki Iwasaki, Masashi Yamazaki

**Affiliations:** 1Department of Orthopaedic Surgery, Ibaraki Prefectural University of Health Sciences Hospital, Ami 300-0331, Japan; mutsuzaki@ipu.ac.jp (H.M.); takeuchir@ipu.ac.jp (R.T.); nakagawasho@tsukuba-seikei.jp (S.N.); 2Center for Medical Sciences, Ibaraki Prefectural University of Health Sciences, Ami 300-0394, Japan; iwasakin@ipu.ac.jp; 3Department of Orthopaedic Surgery, University of Tsukuba, Tsukuba 305-8575, Japan; hkamada@md.tsukuba.ac.jp (H.K.); masashiy@md.tsukuba.ac.jp (M.Y.); 4Department of Physical Therapy, Ibaraki Prefectural University of Health Sciences Hospital, Ami 300-0331, Japan; yoshikawak@ami.ipu.ac.jp (K.Y.); takahashik@ami.ipu.ac.jp (K.T.); 5Department of Physical Therapy, Ibaraki Prefectural University of Health Sciences, Ami 300-0394, Japan; kurodama@ipu.ac.jp; 6Department of Pediatrics, Ibaraki Prefectural University of Health Sciences Hospital, Ami 300-0331, Japan

**Keywords:** robot-assisted gait training, Hybrid Assistive Limb, cerebral palsy, neurologic gait disorders, gait analysis, gait symmetry, walking

## Abstract

*Background and objectives:* Cerebral palsy (CP) is the most frequent childhood motor disability. Achieving ambulation or standing in children with CP has been a major goal of physical therapy. Recently, robot-assisted gait training using the Hybrid Assistive Limb^®^ (HAL) has been effective in improving walking ability in patients with CP. However, previous studies have not examined in detail the changes in gait pattern after HAL training for patients with spastic CP, including gait symmetry. This study aimed to evaluate the immediate effect of HAL training on the walking ability and the changes in gait pattern and gait symmetry in patients with spastic CP. *Materials and Methods*: We recruited 19 patients with spastic CP (13 male and six female; mean age, 15.7 years). Functional ambulation was assessed using the 10-Meter Walk Test and gait analysis in the sagittal plane before and after a single 20-min HAL intervention session. *Results:* The walking speed and stride length significantly increased after HAL intervention compared to the pre-intervention values. Two-dimensional gait analysis showed improvement in equinus gait, increase in the flexion angle of the swing phase in the knee and hip joints, and improvement in gait symmetry. Immediate improvements in the walking ability and gait pattern were noted after HAL training in patients with spastic CP. *Conclusions:* The symmetry of the joint angle of the lower limb, including the trunk, accounts for the improvement in walking ability after HAL therapy.

## 1. Introduction

Cerebral palsy (CP) is the most frequent motor disability in childhood. It is a group of developmental disorders of movement and posture, leading to limitations in activity [1]. CP is associated with the abnormal generation of bioelectric signals in the lower limbs due to brain damage. Motor abnormalities in CP are classified according to the type of tone or movement abnormalities, such as spasticity, dystonia, choreoathetosis, or ataxia. The spastic type accounts for the highest proportion (77–93%) of CP cases [2].

Patients with CP have a limited range of motion due to abnormal motor behavior, including spasticity, resulting in contractures and abnormal gait performance [1,2]. In addition, these features are often asymmetrical, with differences in both sides of the body regarding spasticity and gait dynamics.

Maintenance and improvement of ambulation and standing ability are important features of rehabilitation in patients with CP for improving physical function and for enhancing social interaction [3]. Body weight-supported treadmill training is a type of gait training designed for patients with CP who have high Gross Motor Function Measure scores [4]. In this type of training, the gait speed is based on the participants’ self-selected comfortable speed [5]; despite this advantage, a normal gait pattern is difficult to replicate.

Recently, several studies have described the efficacy of robot-assisted gait training (RAGT) in children with CP [6,7,8,9,10,11]. The representative robots include The Lokomat^®^ (Hocoma, Volketswil, Switzerland), ReoAmbulator^®^ (Motorika, Mount Laurel, NJ, USA) [12], and the Gait Trainer GT1^®^ (Biodex, New York, NY, USA) [13]. Patients who underwent robotic-assisted treadmill therapy showed improvements in their ability to perform functional tasks, such as standing and walking [6,7,8,9,10,11]. Repeated active movement training aligns with the motor learning theory, which is currently popular in physical therapy, as a mode of inducing neuroplastic changes in the brain [14]. These robots enable autonomous motion based on the desired kinematic trajectory of the lower limb joints, mimicking the walking motion of an able-bodied individual.

The Hybrid Assistive Limb^®^ (HAL; Cyberdyne, Tsukuba, Japan) is a wearable robot suit that assists in the voluntary control of knee and hip joint motions [15]. It uses information obtained from the built-in angle, force-pressure and trunk angle sensors, and biopotential signal information obtained through electrodes attached to the wearer’s skin surface. The movement support is determined based on the obtained information, and the power unit arranged in each joint is driven to assist the wearer’s lower limb joint movement. It differs from other robots, as it provides motion according to the wearer’s voluntary drive. HAL training has been reported to improve walking ability and balance in patients with acute [16,17], sub-acute [18], and chronic stroke [19,20], and in those with spinal cord injury [21].

Taketomi and Sankai [22,23] reported that the HAL provides effective walking and stair ascent assistance in patients with spastic CP. We reported that a single gait training session using the HAL, in patients with CP, is safe and can produce immediate effects on walking ability [24]. Additionally, this training enabled immediate improvement in single-leg support and in the hip and knee joint angles during walking [25]. In a representative case, three-dimensional (3D) motion analysis was performed; the flexion angle of the knees decreased at the initial contact and during the late stance phase [26]. We have also reported immediate effects on walking ability and joint angle for the smaller size 2S-HAL training [27]. HAL training in patients with CP has been recognized for its immediate effect by multifaceted evaluation regardless of the measurement method and HAL size.

To date, there are no reported studies examining the HAL training effect on gait pattern, including gait symmetry, in patients with spastic CP. Therefore, this study aimed to examine the immediate effect of HAL training on the walking ability, gait pattern, and gait symmetry in such patients. We hypothesized that changes in gait pattern and symmetry might explain the improvement in walking ability observed in patients with spastic CP who undergo HAL training.

## 2. Materials and Methods

### 2.1. Participants

In total, 19 patients with spastic CP were recruited from the Ibaraki Prefectural Health Science University Hospital from February 2016 to November 2017 (13 male and six female patients; mean age, 15.7 years; range, 9–29 years; mean height, 148.3 cm; range, 131–168 cm; mean weight, 38.7 kg; range, 23–48 kg; Table 1). According to the Gross Motor Function Classification System (GMFCS), three, two, nine, and five participants were classified as level I, II, III, and IV patients. The study protocol was approved by the Ethics Committee of Ibaraki Prefectural University of Health Sciences (approval code (date): 682(14 December 2015), e83(22 December 2016), e119(5 October 2018)). Before participating in the study, all patients and their respective families provided written informed consent.

### 2.2. HAL Protocol

For this clinical study, the patients used a HAL size S (target 145–165 cm, weighing approximately 14 kg). A walking device (All-in-One Walking Trainer; Healthcare Lifting Specialist, Odense, Denmark) with a harness was used for safety purposes, and the HAL intervention consisted of walking with the assistance of two physical therapists (Figure 1). Each patient underwent a single session of HAL training. The session lasted 60 min, including a period of rest, an evaluation period before and after the intervention, and time for attachment/detachment of the device (10, 10, and 20 min, respectively). The actual training time using the HAL was 20 min. Before starting the training, the patients’ height, body weight, hip angle, knee angle, and foot deformation were recorded. For patients whose joint movement range was severely restricted, limits were placed for the HAL range of motion. The cybernetic voluntary control mode was used. In addition, flexion/elongation balance and torque assistance in the hip and knee joints were optimized for each patient. The thigh and shank length and the hip width of the HAL were adjusted accordingly. 

### 2.3. Outcome Measures

Functional ambulation was assessed before and after the intervention, without wearing the HAL; walking speed and cadence were measured using the 10-m Walk Test (10MWT) and a video was recorded on the sagittal plane and analyzed. We analyzed one gait cycle of each patient with Dartfish Team Pro version 5.5 (Dartfish, Fribourg, Switzerland). The video was played in slow motion, and pauses were made to measure the stride length and the angles of the hip, knee, ankle, and trunk at the beginning of each gait cycle phase. Video images were played back frame-by-frame to determine the beginning of each gait cycle. A gait cycle was defined as the movement starting from initial contact on one side until the next initial contact on the same side. The beginning of each phase of the gait cycle was defined as follows: in the stance phase, the loading response began with the initial contact, mid-stance with the toe-off on the opposite side, and terminal stance with the heel-off; in the swing phase, there was pre-swing with the initial contact of the opposite side, initial swing with the toe-off, mid-swing with the intersection of the feet, and terminal stance when the lower leg was vertical.

Statistical analyses were performed using the Statistical Package for Social Science (SPSS) version 22.0 (IBM Corp., Armonk, NY, USA). We compared the outcome measures before and after the training using paired *t*-tests. The results were analyzed and compared using average values; a two-tailed *p*-value < 0.05 was considered as statistically significant. To evaluate left-right symmetry, the change in the difference between the left and right joint angles was examined using Cohen’s d. A change in the effect size >0.2 was considered as an improvement in symmetry. 

## 3. Results

Two patients (patients no. 15 and 19) could not walk and did not have a gait assistive device. Patients no 9 and 17 were lifted by the therapist during gait, while the first patient sat on the walker’s saddle during gait. Patients with GMFCS level IV could perform HAL training with more support from the therapists, but their gait performance could not be evaluated. Gait parameters and video analyses were performed only in patients with GMFCS levels I–III (*n* = 14). No adverse events were observed in any of the patients using HAL.

The results of walking ability analysis are summarized in Table 2. The gait performance of patient no. 5 was evaluated, but the patient could not perform the 10MWT because he completed the procedure using parallel bars. HAL training increased the walking speed (*p* = 0.003), right stride length (*p* = 0.021), and left stride (*p* = 0.014) performances. There were no differences between left and right stride improvements.

The joint angles before and after HAL training and the results of left-right joint angle symmetry are presented in Figure 2 and Figure 3, respectively. In patients with equinus gait, it was impossible to define the terminal stance because they had no moment of heel-off. Meanwhile, we were not able to define the terminal swing in patients with crouching gait because they had no movement of the vertical lower leg. In patients with asymmetric gait, the definable gait cycle was different on the left and right sides; thus, the t-test of the lateral difference of each joint could not be performed. Instead, the effect size of Cohen’s d was used to compare the left-right symmetry.

After HAL training, significant improvements were noted in the anterior tilt angle of the right trunk at the terminal swing (Figure 2a,b), flexion angle of the left hip joint at mid-swing (Figure 2c,d), flexion angle of the right knee joint at the terminal stance (Figure 2e,f), and plantar flexion angle of the right ankle joint at mid-stance (Figure 2g,h).

Concerning joint-angle symmetry, improvements were noted in the following joints: the trunk at the terminal stance and terminal swing (Figure 3a,b); hip at the mid-swing (Figure 3c,d); knee at the mid-stance, terminal stance, mid-swing, and terminal swing (Figure 3e,f); and ankle at the loading response, mid-stance, terminal stance, initial swing, mid-swing, and terminal swing (Figure 3g,h). All these changes presented a statistical significance. Left-right symmetry improvements were noticed in the gait cycle phases where the range of joint motion also changed significantly, as depicted in Figure 3. The change in the joint angle was not so significant, but in some phases, there was improved symmetry. Ankle at the initial swing, the right dorsiflexion angle became smaller and the left became larger, so the difference between the left and right increased and the joint-angle symmetry deteriorated. It was considered that the influence of orthosis was involved in the ankle joint angle.

The characteristics of gait cycles before and after RAGT using HAL are shown in Figure 4. There was no significant difference in the percentage of the gait cycle (Figure 4a). After the intervention, the double-support period was significantly shorter, and the swing phase was longer, respectively, compared to the baseline measurements (Figure 4b); the total gait cycle time decreased for both sides, but this change was not statistically significant (Figure 4a).

## 4. Discussion

In this study, we examined the immediate effect of HAL training on the walking ability and gait pattern of patients with spastic CP. We found that improvements in the equinus gait and increases in the flexion angle of the swing phase in the hip joints resulted in increased walking speed, extended stride, and improved symmetry (Figure 5). The HAL-induced joint symmetry and gait pattern accounted for the improved walking ability, thus, confirming our hypothesis.

We had previously reported that a single session of HAL training in patients with CP immediately improved walking speed and stride [25], increased hip and knee joint angles during the stance and swing phases, and enhanced single-leg support per gait cycle [24]. In this study, we measured all joint angles of the lower limbs, including the trunk angle. Apart from confirming the previously documented increase in joint angles, we also revealed that the trunk and lower limb joint symmetries contributed to gait improvement. In cases where a single intervention was performed twice, the walking ability was maintained, and further improvement was observed after the second training.

HAL can potentially intensify feedback on muscle activity during rehabilitation. HAL-induced motion might also evoke sensory input, which has a favorable feedback effect on the central nervous system and plays an important role in the recovery of locomotor function [21]. It has been reported that in a patient with cerebral palsy, improved activity of the vastus lateralis and semitendinosus muscles during gait was attributed to neurophysiological changes induced by HAL training [28]. Interestingly, HAL training in patients with CP reduces co-contraction and expands the passive joint range [26,27]. Therefore, it will be necessary to evaluate the symmetry using an electromyogram.

Using this device, patients with CP can perform a voluntary gait that is close to normal and, therefore, its feedback effects are expected to result in substantial improvements in walking ability.

As GMFCS level III and IV patients have strong motor difficulties, it becomes difficult to evaluate their gait performance [25]. In this study, GMFCS level IV patients undergoing HAL training mainly performed standing exercises. In these cases, the standing motion should have been assessed, but this was not our purpose.

This study had several limitations. First, the sample size was small, and our analysis focused on the immediate effects of a sole session of HAL training. Second, although video-assisted gait analysis with Dartfish allowed us to accurately determine the phases of the gait cycle and the joint angles [29], it did not permit a sufficient evaluation of valgus, adduction, and scissor walking; to compare these features, 3D motion analysis is necessary. Third, this study evaluated the immediate effect and did not include a long term follow up. 

Further studies are needed to determine if a progressive period of training significantly improves pattern and gait symmetry. Interestingly, in a previous case report where the single intervention was performed twice, the walking ability was maintained [30]. We reported that 12 HAL training sessions improved the walking speed and cadence in six patients with CP [31]; we speculated that increasing the number of HAL training sessions could provide further benefit on gait. Future studies including a larger number of patients should be conducted to confirm our findings and should include multiple HAL training sessions and their long-term follow up in patients with CP.

## 5. Conclusions

A single HAL training session improved walking speed and step length in patients with spastic CP. In addition, corrected joint angle symmetry seems to play a significant role in the walking abilities of patients with CP.

## Figures and Tables

**Figure 1 medicina-56-00673-f001:**
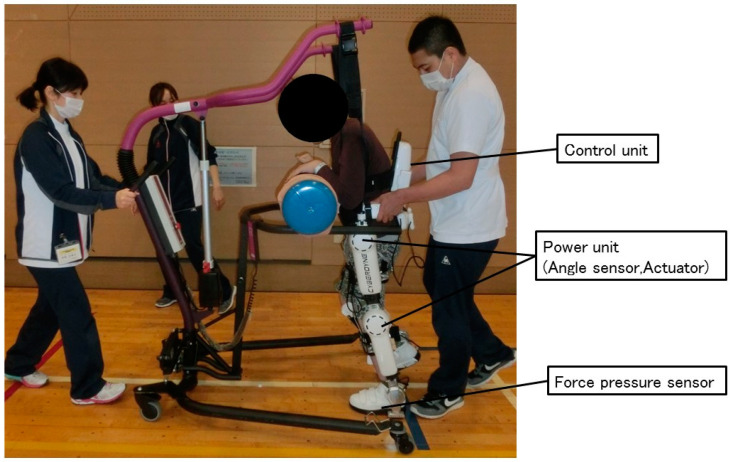
Robot-assisted gait training using the Hybrid Assistive Limb (HAL). A walking device with a harness is used, and two physical therapists assist the patient.

**Figure 2 medicina-56-00673-f002:**
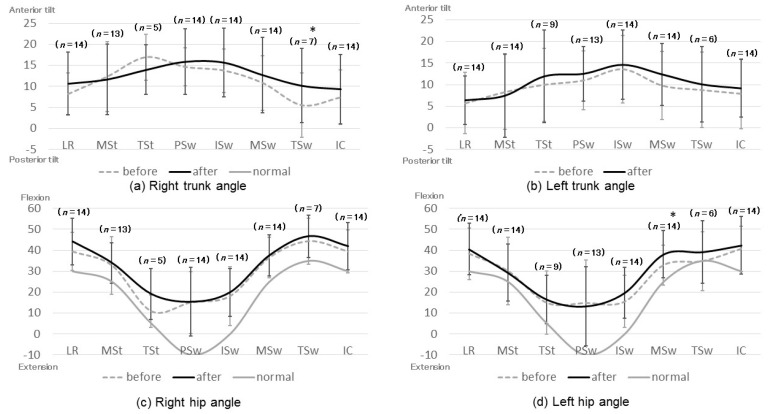
Joint angles before and after HAL training. All with respect to the horizontal line. * Joint angles that increased significantly after HAL training. LR: loading response; MSt: mid-stance; TSt: terminal stance; PSw: pre-swing; ISw: initial swing; MSw: mid-swing; TSw: terminal swing; IC: initial contact.

**Figure 3 medicina-56-00673-f003:**
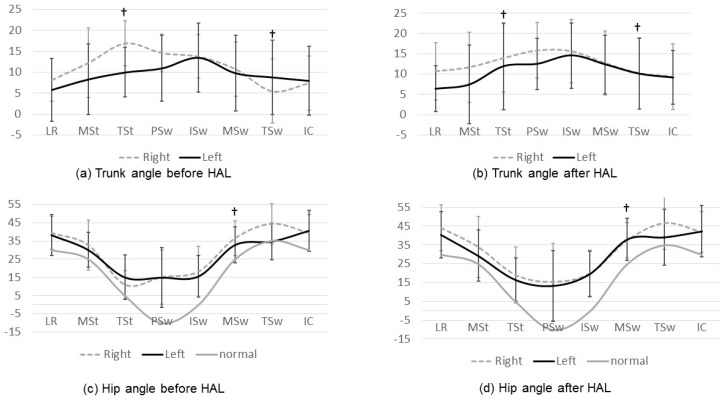
Left-right symmetry in the joint angles. The left-right symmetry improved in the phases where a significant difference in the joint angle was also noticed. †: Left-right symmetry significantly improved with HAL training.

**Figure 4 medicina-56-00673-f004:**
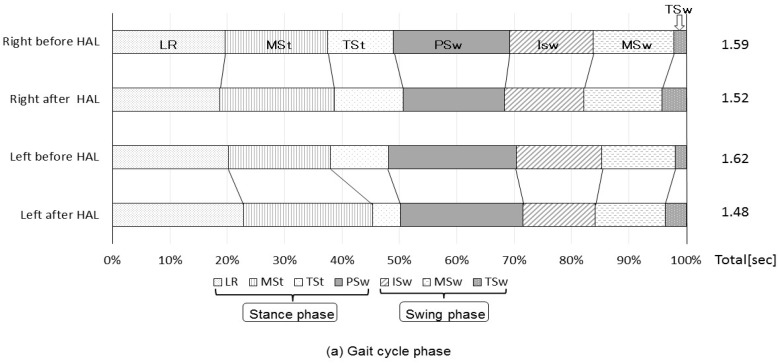
Changes in gait cycle phases. The ratio of each phase to the gait cycle duration is shown before and after HAL training. *: The percentage of phase significantly changed after HAL training.

**Figure 5 medicina-56-00673-f005:**
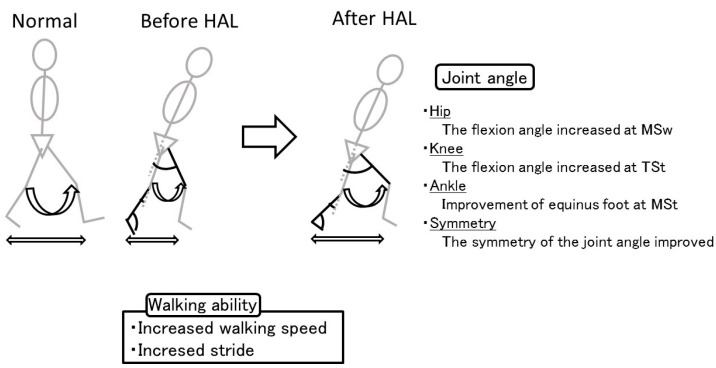
Gait pattern. The improvements in equinus gait and flexion angle of the swing phase in the hip joints increased the walking speed, extended the stride, and improved the symmetry.

**Table 1 medicina-56-00673-t001:** Participants’ characteristics.

Case	Sex	Age (Years)	Height (cm)	Weight (kg)	Paralysis	GMFCS	Assistive Device	Orthosis	10MWT	Gait Evaluation
1	F	19	152	40	Diplegia	II	NA	NA	+	+
2	M	12	138	27	Diplegia	III	Walker	AFO	+	+
3	M	16	153	49	Diplegia	III	Walker	AFO	+	+
4	F	11	137	35	Diplegia	III	Walker	AFO	+	+
5	M	20	168	64	Diplegia	III	Parallel bars	AFO	-	+
6	F	15	141	48	Diplegia	III	Walker	AFO	+	+
7	M	15	158	43	Diplegia	III	Walker	AFO	+	+
8	F	15	143	35	Diplegia	II	Crutch	AFO	+	+
9	M	14	153	37	Diplegia	IV	BS walker	AFO	-	-
10	M	11	132	32	R hemiplegia	I	NA	R AFO	+	+
11	M	9	134	28	R hemiplegia	I	NA	R AFO	+	+
12	M	10	134	36	Diplegia	III	NA	AFO	+	+
13	F	12	153	24	Quadriplegia	IV	BS walker	AFO	-	-
14	M	23	165	45	Diplegia	III	Crutch	AFO	+	+
15	M	16	152	35	Diplegia	IV	-	AFO	-	-
16	M	13	150	36	R hemiplegia	I	NA	NA	+	+
17	M	10	146	38	Diplegia	IV	Parallel bars	AFO	-	-
18	F	22	155	41	Quadriplegia	III	Crutch	AFO	+	+
19	M	29	165	45	Quadriplegia	IV	-	AFO	-	-

GMFCS: Gross Motor Function Classification System; 10MWT: 10-m Walk Test; F: female; M: male; NA: not applicable; AFO: ankle foot orthosis; BS walker: body support walker; R: right; +: execution; -: not execution.

**Table 2 medicina-56-00673-t002:** Outcome measures before and after HAL training.

Outcome Measurements	Before Using the HAL	After Using the HAL	*p*-Value	*n*
Speed (m/s)	0.78 ± 0.35	0.93 ± 0.43	0.003 *	13
Cadence (steps/min)	101.6 ± 31.1	108.6 ± 42.4	0.223	13
Right stride length (m)	0.92 ± 0.27	1.02 ± 0.28	0.020 *	14
Left stride length (m)	0.93 ± 0.27	1.05 ± 0.28	0.014 *	14
Right/left stride length	0.99 ± 0.14	0.98 ± 0.09	0.411	14

* *p* < 0.05; HAL: Hybrid Assistive Limb.

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
