# Peer review of "Effect of the Hybrid Assistive Limb on the Gait Pattern for Cerebral Palsy"

_medicina, 2020, doi:10.3390/medicina56120673_

Round 1

Reviewer 1 Report

Suggestions

Effect of the Hybrid Assistive Limb on the Gait Pattern for Cerebral Palsy 

Change title 

Effect of a Single Session of the Hybrid Assistive Limb on the Gait Pattern for Cerebral Palsy 

Overall, this is an interesting analysis of change in gait dynamics using a Hybrid Assistive Limb. The major weakness is that it is a single session of intervention for 19 subjects which is an impractical intervention. However, it is interesting to discover that the improvement in symmetry and speed was due to increase in flexion of the knee and hip in swing phase.

The authors tried to address the comments from the first review.

The topic is of interest ( robotic training for CP)

The study is well written and presented with a lot of figures etc. 

I would suggest the authors do some editing: 

  1. Gait speed should be measured in m/second ( not m/minute) 
  2. Provide a stick figure of normal gait 
  3. Provide a figure of normal joint angle curves 
  4. Explain line 175….. “the joint angle symmetry deteriorated  
  5. Improve the sentence for 184 and 185…. It is confusing 
  6. The paragraph on study limitations is weak and should not refer to previous studies. The information on previous studies can be included but not here.  
  7. Add a limitation that the study did not include a long term follow up 
  8. Recommendations for further studies:

a. Add, Further studies are needed to determine if a progressive period of training significantly improves the pattern and gait symmetry. Future studies need to include a long term follow up to look at integration of training over time. Additional studies are also needed to determine if non-walkers with CP can be mobilized to walkers with HAL traiing 

shed with a single intervention in 19 subjects.  The researchers cannot change anything unless they did another study and included multiple treatment sessions. 

Author Response

We thank you for your thoughtful suggestions and insights. The manuscript has benefited from these insightful suggestions.The revisions are denoted in blue font in the revised manuscript and the previous revisions in red font.The responses to all comments have been prepared and provided below.

Thank you for your consideration.

  • Gait speed should be measured in m/second ( not m/minute) 

Response: We have changed the measurement of gait from m/minute to m/second.

  • Provide a stick figure of normal gait 

Response: Thank you for the suggestion. We have added the stick figure of normal gait.

  • Provide a figure of normal joint angle curves 

Response: As per your recommendation, which we agree with, we have added normal joint curves in Figure 2 and 3

  • Explain line 175….. “the joint angle symmetry deteriorated

Response: We apologize for the unclear statement. We have revised the statement for clarity and now reads: “the difference between the left and right increased and the joint-angle symmetry deteriorated.” (Lines 175-176).

  • Improve the sentence for 184 and 185…. It is confusing .

Response: Thank you for the suggestion. We have revised the sentence, which now reads: “the double-support period was significantly shorter, and the swing phase was longer” (Lines 184 and 185).

  • The paragraph on study limitations is weak and should not refer to previous studies. The information on previous studies can be included but not here.  

Response: We appreciate the suggestion. We have moved the citations of the previous studies within the limitation to the next paragraph (Lines from 229 to 231).

  • Add a limitation that the study did not include a long term follow up 

Response: Thank you for the suggestion, which we have effected by including the sentence: “this study evaluated the immediate effect and did not include a long term follow up.” (Lines 227 and 228).

  • Recommendations for further studies:
  • Further studies are needed to determine if a progressive period of training significantly improves the pattern and gait symmetry. Future studies need to include a long term follow up to look at integration of training over time. Additional studies are also needed to determine if non-walkers with CP can be mobilized to walkers with HAL training shed with a single intervention in 19 subjects.  The researchers cannot change anything unless they did another study and included multiple treatment sessions. 

Response: We thank you for the thoughtful suggestions. We have revised the section on further studies to the following sentence: “Further studies are needed to determine if a progressive period of training significantly improves pattern and gait symmetry. Interestingly, in a previous case report where the single intervention was performed twice, the walking ability was maintained [30]. We reported that 12 HAL training sessions improved the walking speed and cadence in six patients with CP [31]; we speculated that increasing the number of HAL training sessions could provide further benefit on gait. Future studies including a larger number of patients should be conducted to confirm our findings and should include multiple HAL training sessions and their long-term follow up in patients with CP.” (Lines from 229 to 235).

Reviewer 2 Report

Minor comments:

Line 133 and 135: space should be placed between < and 0.05 and 0.2.

Line 138: Patient should be changed to Patients.

Author Response

We thank you for your thoughtful suggestions and insights. The manuscript has benefited from these insightful suggestions.The revisions are denoted in blue font in the revised manuscript and the previous revisions in red font. The responses to all comments have been prepared and provided below.

Thank you for your consideration.

Line 133 and 135: space should be placed between < and 0.05 and 0.2.

Response: We have added the spaces as suggested: “p-value < 0.05” “the effect size > 0.2” (Lines 133 and 135).

Line 138: Patient should be changed to Patients.

Response: We have changed Patient to Patients. (Lines 139)

Reviewer 3 Report

The authors sufficiently addressed the revisions for this manuscript.

Author Response

Thank you for your peer reviews.

The manuscript has been rechecked, and the necessary changes have been made in accordance with other reviewers’ suggestions.

Thank you for your consideration.

This manuscript is a resubmission of an earlier submission. The following is a list of the peer review reports and author responses from that submission.

Round 1

Reviewer 1 Report

Hybrid gait training for patients with CP

This topic is of interest in pediatrics and has applications to adults with CP as well as those with neurological injuries.  The topic is of primary interest for those interested in rehabilitation strategies to improve gait using robotic devices in patients with neurological impairments. 

A significant weakness is reporting the outcomes from a single intervention of hybrid, robotic gait retraining.  The impact of a single session of retraining in patients with chronic neurological impairments provides no insight into long term improvement in gait symmetry, gait stability, gait speed or gait endurance.  It is disappointment to me that the researchers had a  good group of 19  patients with CP ( mostly adults with CP) and did not study the impact of repetitive hybrid robotic gait training. 

The study is clearly reported with good supporting data .  In terms of statistics, the authors need to clarify if the testing was two tailed at p<0.05 or one tailed. 

The conclusion was appropriately limited to the one session of training

“A single session of HAL training improved walking speed and step length in patients with spastic CP. In addition, corrected joint angle symmetry seems to play a significant role in the walking abilities of patients with CP.”

 The question is whether this one session effect can be translated to  a repetitive training study associated with long term improvement in gait speed, safety, symmetry and quality.

Based on previous studies by the same researchers, the only additional piece of information gained with this study was noting  improvement in trunk and joint symmetries which may have contributed to improved gait. 

How does one know if the improvement in gait of a single session  or robotic training involving 2 physical therapists and a researcher  was due to the placebo effect rather than a direct effect of 20 minutes of training with a hybrid robotic device.  

Reviewer 2 Report

The authors hypothesize that the improvement of the walking ability of spastic CP who undergo HAL training can be due to the changes in the gain pattern and symmetry. They have recruited 19 spastic CP. The functional ambulation of the participants is assessed using 10-Meter Walk Test and gait analysis in the sagittal plane before and after a single 20 minutes HAL intervention session. The authors report an increase in the walking speed and stride length after HAL intervention compared to the pre-intervention values. Equinus gait improvement, an increase of the flexion angle of the swing phase in the knee and hip joints, and gait symmetry improvement of spastic CP participants are reported as well. Compared to the authors' previous studies, they have included the measurement of all joint angles of the lower limbs including the trunk angle.

Comments:

  1. Line129: d-value of 0.15 is chosen for the statistical analysis of Cohen's d. However, d-value of 0.2 is considered as 'small' effect size. Why a value of 0.15 is adopted?
  2. Spasticy is one of the sequels of the brain damage of the CP. In this study, you also discuss the spasticity (e.g. line 178). However, you have not investigated spasticy in this study. How did the spasticity of the CP change before and after HAL invervention session? Don't you think that spasticty is an important factor which should be investigated in these kinds of studies?

Reviewer 3 Report

The manuscript is generally well-written with some places of awkwardness, i.e.p. 2 line 62 physical treatment as a mode of inducing...treatment is awkward.  The manuscript should be checked for patient-first language i.e. p. 8 line 197 "CP patients" should be patients with CP

Major concerns:

  1. The introduction would benefit having more information about HAL-how does it differ from the other robot-assisted gait devices?
  2. The methods would also benefit having more information about HAL-how is a signal detected by the wearer's nervous system, how heavy it is to wear?
  3. Why not remove cases 9, 13, 17 and 19 from Table 1 and just report their result) ambulation required assistance of two, unable to perform 10MWT or gait evaluation, etc)?
  4. In Table 2, the n varies between 13 and 14. Need to explain why.
  5. In the discussion, the authors fail to explain the results compared to normal gait and to the work of others. There is no explanation for the discrepancy between the result for the right and left joint angles (Figure 2), nor the left-right symmetry result (Figure 3).
  6. I would think that reporting the average result of an n of 13/14 is a limitation of the study.
  7. Minor concerns:
    1. Table 1-the symbol + is not explained. Why not just use NA for the last two columns?
    2. Are cases 15 & 19 ambulatory?
    3. Figure 4-the key is difficult to read because of the small font size. Suggest label the X axis with each phase of gait instead.
    4. The sentence ending on line 43 of p.2 needs a reference.